# Upper Respiratory Infection Drives Clinical Signs and Inflammatory Responses Following Heterologous Challenge of SARS-CoV-2 Variants of Concern in K18 Mice

**DOI:** 10.3390/v15040946

**Published:** 2023-04-11

**Authors:** Jacob H. Nichols, Evan P. Williams, Jyothi Parvathareddy, Xueyuan Cao, Ying Kong, Elizabeth Fitzpatrick, Richard J. Webby, Colleen B. Jonsson

**Affiliations:** 1Department of Microbiology, Immunology and Biochemistry, College of Medicine, University of Tennessee Health Science Center, Memphis, TN 38163, USA; jnicho60@uthsc.edu (J.H.N.); ewilli99@uthsc.edu (E.P.W.);; 2Regional Biocontainment Laboratory, University of Tennessee Health Science Center, Memphis, TN 38163, USA; 3Department of Health Promotion and Disease Prevention, College of Nursing, University of Tennessee Health Science Center, Memphis, TN 38163, USA; 4St. Jude Children’s Research Hospital, Memphis, TN 38163, USA; 5Institute for the Study of Host-Pathogen Systems, University of Tennessee Health Science Center, Memphis, TN 38163, USA

**Keywords:** SARS-CoV-2, infection, challenge, reinfection, K18-hACE2 mouse, variants of concern, RNASeq, Alpha variant, Delta variant

## Abstract

The evolution of the severe acute respiratory syndrome coronavirus 2 (SARS-CoV-2) has resulted in the emergence of several variants of concern (VOC) with increased immune evasion and transmissibility. This has motivated studies to assess protection conferred by earlier strains following infection or vaccination to each new VOC. We hypothesized that while NAbs play a major role in protection against infection and disease, a heterologous reinfection or challenge may gain a foothold in the upper respiratory tract (URT) and result in a self-limited viral infection accompanied by an inflammatory response. To test this hypothesis, we infected K18-hACE2 mice with SARS-CoV-2 USA-WA1/2020 (WA1) and, after 24 days, challenged with WA1, Alpha, or Delta. While NAb titers against each virus were similar across all cohorts prior to challenge, the mice challenged with Alpha and Delta showed weight loss and upregulation of proinflammatory cytokines in the URT and lower RT (LRT). Mice challenged with WA1 showed complete protection. We noted increased levels of viral RNA transcripts only in the URT of mice challenged with Alpha and Delta. In conclusion, our results suggested self-limiting breakthrough infections of Alpha or Delta in the URT, which correlated with clinical signs and a significant inflammatory response in mice.

## 1. Introduction

The rate of mutation of SARS-CoV-2 coupled with its spread globally in a genetically diverse human population has resulted in hundreds of variants. However, only three have driven global infection, the variants of concern (VOCs) of Alpha, Delta, and Omicron. The early surge in coronavirus disease 2019 (COVID-19) cases was dominated by the rise in the prevalence of strains containing spike mutations, e.g., D614G and N501Y, which improved viral transmission by increasing the spike’s affinity for human angiotensin, converting enzyme-2 (ACE2), the primary receptor for SARS-CoV-2 [1,2]. The Alpha, Delta, and Omicron variants followed in successive waves, each with mutations that improved their ability to evade the host immune response and transmit [3,4]. Current vaccination strategies employ only the spike glycoprotein as the antigen, and since the VOCs have accumulated numerous mutations in the receptor binding domain [5], it is important to know how these mutations might impact vaccine efficacy. For example, neutralizing antibodies (NAbs) generated in response to immunization with the first generation of vaccines have notable drops in neutralizing efficacy in the months after immunization [6]. A third booster dose proved necessary to restore detectable humoral responses against Omicron, and the necessity of additional rounds of boosters is currently being evaluated [7]. While we know a great deal about the efficacy of the SARS-CoV-2 vaccines [8], less is known about the protective efficacy from prior natural infection. The current mRNA vaccines have a proven ability to generate more effective neutralizing antibodies than natural infection alone [9]. However, there is evidence that individuals who had been both immunized with an mRNA vaccine and naturally infected with SARS-CoV-2 have a more robust neutralizing antibody response to VOCs than those vaccinated with two doses or naturally infected [10,11,12,13]. As the virus continues to proliferate globally, more data is necessary to fully appreciate the effects of natural SARS-CoV-2 infection on immunity.

To study the host response to homologous and heterologous reinfection with SARS-CoV-2, we infected K18-hACE2 mice with the WA1/2020 (WA1) strain and then challenged with a high dose of WA1, Alpha, or Delta three weeks later. We report that an infection with a low dose of SARS-CoV-2 confers significant protection from challenge, but mice in the heterologous challenge groups had greater weight loss and a heightened proinflammatory response in the nasal turbinates. Moreover, despite the differences in clinical signs and host response between homologous- and heterologous-challenged mice, there was no significant difference among challenge groups in the fold change of their 50 percent plaque reduction neutralization titers (PRNT_50_) to any of the tested antigens three weeks after challenge.

## 2. Materials and Methods

### 2.1. Cells, Viruses, and Next-Generation Sequencing

All reagents and cell culture reagents were purchased from Thermo Fisher Scientific unless specified. Vero E6 (ATCC CRL-1586) and Vero TMPRSS2 [14] were maintained in Minimum Essential Medium (MEM) with Earle’s salts with L-glutamine (Corning), 1% penicillin/streptomycin (Gibco) and 10% heat-inactivated fetal bovine serum (FBS) (Gibco) (complete MEM). The SARS-CoV-2 USA-WA1/2020 (WA1) was obtained through BEI Resources Centers for Disease Control and Prevention GenBank accession no. NR-52281. The Alpha strain, B.1.1.7 (hCoV-19/USA/TN-UT2590/2021, UTHSC ID UT2590, GenBank accession no. OP628428), and the Delta strain, B.1.617.2 AY122, (hCoV-19/USA/COR-21-160048/2021, St. Jude ID C0517, GenBank accession no. OP677756), were isolated from nasal swab samples, plaque-isolated, amplified, and sequenced as described previously [15]. Acquisition of deidentified nasal pharyngeal swabs was determined not to meet the definition of human subject research (University of Tennessee Health Science Center (UTHSC) 20-07254-NHSR). Virus seed stocks were amplified in Vero E6 TMPRSS cells in DMEM with 2% FBS and 1 mg/mL Geneticin and measured by plaque assay as described [15]. Cells were not used beyond passage 30. Virus seed stocks were frozen in 0.5 mL aliquots and kept at −80 °C until used. All experiments using infectious viruses were conducted within the BSL-3 or ABSL-3 areas of the UTHSC Regional Biocontainment Laboratory (RBL).

We used next-generation sequencing to confirm the consensus sequence of viral seed stocks. Viral RNA was extracted using MagMAX™ Viral/Pathogen Nucleic Acid Isolation Kit (Applied Biosystems, Waltham, MA, USA) on KingFisher Flex instrument (Thermo Fisher). cDNA was synthesized using SuperScript IV First-Strand Synthesis System (Invitrogen). cDNA was amplified by using Q5 High Fidelity polymerase (Hot Start High-Fidelity 2X Master Mix, New England Biolab) and ARTIC version 3 nCoV-2019 Amplicon Panel (Integrated DNA Technologies, Coralville, IA, USA). Sequencing libraries were prepared from PCR products using Nextera XT DNA Library Prep Kit (Illumina, San Diego, CA, USA) and sequenced on a MiSeq instrument using MiSeq Reagent Kit v3 (150-cycle) (Illumina).

### 2.2. General Animal Information and Study Design

Six-week-old female K18-hACE2 mice (Jackson Laboratory, Bar Harbor, ME, USA) were acclimated in the UTHSC RBL ABSL-2 in Allentown BCU cages. Mice were provided solid food and autoclaved water, along with irradiated gel food and water cups after the first day of infection. Bio Medic Data Systems IPTT-3000 transponder tags were inserted subcutaneously to uniquely identify each mouse in the study and monitor temperature. Digital and written logs of each mouse’s weight and temperature were taken each morning. On the day of infection, mice were transported to the RBL ABSL-3. For intranasal infection, mice were anesthetized by isoflurane and were inoculated intranasally with either a 50 µL (25 µL/nare) solution containing 500 plaque-forming units (PFU) of WA1 (n = 64) or 50 µL (25 µL/nare) of Dulbecco’s Phosphate Buffered Saline (PBS) (n = 40). The virus was diluted to 500 PFU in 50 µL PBS prior to infection. One hundred and eight mice were weighed, and temperatures were taken prior to infection at 0 days postinfection (dpi), and this continued each day throughout the study. The clinical signs were monitored twice per day until recovery and then once daily after recovery. Clinical signs scored included lethargy based on appearance, respiratory effort, weight loss, body temperature reduction, behavior (activity), and hydration status.

Four mice from the WA1-infected group were euthanized at 3 dpi. At 21 dpi, blood was collected retro-orbitally from anesthetized mice to measure neutralizing antibody titers by PRNT_50_. At 24 dpi, mice were anesthetized with isoflurane and infected with a total of 50 µL (25 µL/nare) of 2.5 × 10^4^ PFU of WA1 (n = 12), Alpha (n = 12), Delta (n = 12), or PBS (n = 8). Lungs and nasal turbinates (n = 4 from each group) were taken at 25 dpi and 27 dpi; 4 mice per group remained for survival. During necropsy, lungs, and nasal turbinates were immediately homogenized in Beadmill tubes containing 2.8 mm ceramic beads with 1 mL of lysis buffer (MagMAX mirVana^TM^ Total RNA Isolation Kit lysis buffer). At 45 dpi, blood was collected by cardiac puncture, and mice were euthanized by isoflurane and cervical dislocation. To separate serum, blood was centrifuged at 3000× *g* for 10 min in EDTA tubes and transferred to cryovial tubes. All sera and tissues were immediately flash frozen in liquid nitrogen and stored at −80 °C. Studies were conducted in accordance with and approved by the Institutional Animal Care and Use Committee of the UTHSC (Protocol #20-0132).

### 2.3. RNA-Seq

Total RNA was isolated from each homogenized lung and nasal turbinate sample according to the MagMAX mirVana protocol using a KingFisher Flex instrument (Thermo Fisher). The quality and quantity of lung and nasal turbinate isolated total RNA were measured on an Agilent Fragment Analyzer 5300. RNA-Seq libraries were prepared from isolated total RNA using NEBNext^®^ Ultra II Directional RNA Library Prep Kit for Illumina (NEB). Completed libraries were measured on the Fragment Analyzer 5300 and sent to Novogene for sequencing on Illumina’s NovaSeq platform (2 × 150, 60 M pair-end reads per sample).

### 2.4. RT-qPCR

Total RNA was isolated from each homogenized lung and nasal turbinate sample according to the MagMax mirVana protocol using a KingFisher Flex instrument. RT-qPCR was performed using the Luna SARS-CoV-2 RT-qPCR Multiplex Assay Kit (New England Biosciences) on a QuantStudio 6 Flex System (ThermoFisher). Viral RNA was quantified in reference to a standard curve (R^2^ = 0.9968) that was generated based on 10-fold dilutions of RNA isolated from WA1 stock of a known concentration (PFU/mL). Standard curves were generated and evaluated using GraphPad Prism version 9.5.0.

### 2.5. Plaque Reduction Neutralization Test

One hundred µL of diluted virus (75 PFU) was added to 100 µL of sera prepared with 7 2-fold dilutions (1:100–1:12,800) in duplicate and incubated for 1 h at 37 °C and 5% CO_2_. The mixture was used to infect 12-well plates seeded with 2.5 × 10^5^ Vero E6 cells made the day prior. Next, a 50/50 overlay mixture of 2% CMC and Modified Eagle Medium (2X) supplemented with 10% HI-FBS and 1% P/S was added, and the cells were incubated at 37 °C and 5% CO_2_ for 3 days. Cells were fixed with 10% formalin for 30 min and stained with 1% crystal violet in a 10% glacial acetic acid solution for 10 min to visualize plaques for counting. Plaque counts were recorded and run through an in-house R script with Four Parameter Logistic model to generate PRNT_50_ data. The R script is available on request.

### 2.6. Bioinformatics

Sequencing read results were processed on CLC Genomics Workbench Version 22 (Qiagen, Hilden, Germany). Each sample underwent trimming and filtering using default parameters followed by mapping sequencing reads to the concatenated genome of Mus musculus (GRCm39.105 with the reference sequence of the viruses stated above). Gene count results were used to calculate differential gene expression by DESeq2 [16], and genes that exhibited a log_2_ fold change ≥ 1.5 and a false discovery rate ≤ 0.05 were used to assess canonical pathways using Ingenuity Pathway Analysis (Qiagen) [17]. SARS-CoV-2 genome depth of coverage was calculated by SAMtools Version 1.13.

### 2.7. Statistics

Graphs were generated through GraphPad Prism 9.3.1. Ordinary one-way ANOVA was used to generate calculations of *p*-value significance in the percent weight change graph between WA1:WA1, WA1:Alpha, and WA1:Delta group graphs. The fold changes of PRNT_50_ between 21 dpc and 21 dpi were log-transformed. A 1-sided 1-sample *t*-test was used to test mean log-transformed fold changes greater than 0. We used a 2-sample *t*-test to compare the difference in fold changes between the two groups. A robust linear regression was used to model fold change (log-transformed) with rechallenge and antigen as predictors. All tests are two-sided unless specified and implemented in R version 4.2.1.

## 3. Results

### 3.1. A Low Dose of SARS-CoV-2 Confers Greater Protection in Homologous- versus Heterologous-Challenged Mice

As presented in the study timeline (Figure 1A), mice were inoculated with PBS or infected with a low dose (500 PFU) of the A lineage strain, WA1. The dose was chosen to enable infection but allow at least two-thirds of the cohort to survive for the subsequent challenge reported previously [18]. Of the low-dose WA1-infected cohort, 44/64 (68.75%) survived infection (Figure 1B). Mice infected with the low dose of WA1 showed a drop in weight starting at 4 dpi that continued through 7 dpi, at which point most began to recover until they regained 100% of their original weight by 10 dpi (Figure 1C).

Twenty-four days postinfection (dpi) or 0 days postchallenge (dpc), mice were challenged with a high dose (2.5 × 10^4^ PFU) of WA1, Alpha (B.1.1.7), Delta (B.1.617.2), or PBS (Figure 1A). Following challenge at 24 dpi, the average percent weight change for WA1-infected:WA1-challenged (WA1:WA1) mice trended with PBS-inoculated mice and continued to increase (Figure 1C). In contrast, theWA1:Alpha and WA1:Delta showed a weight decrease on days 24–28. The average weight is shown for clarity (Figure 1C). Evaluation of this initial weight drop suggests a mild illness in WA1:Alpha and WA1:Delta. Outside of the drop in weight for the WA1:Alpha and WA1:Delta groups, no other clinical symptoms were observed, and none of the WA1:Alpha or WA1:Delta mice reached the criteria for euthanasia. The clinical signs monitored included lethargy based on appearance, respiratory effort, weight loss, body temperature reduction, behavior (activity), and hydration status.

### 3.2. WA1, Alpha, and Delta Showed a Similar Boost in Neutralization Titer

To evaluate the neutralization titers prior to and after challenge, blood was drawn at 21 dpi (−3 dpc) and 45 dpi (21 dpc). The fold change of the 50 percent plaque reduction neutralization titers (PRNT_50_) for sera collected at 21 dpi (Figure 2A) and 45 dpi (21 dpc) were compared for mice in each experimental group using WA1, Alpha, or Delta as antigen (Figure 2B, Appendix A). Analyses of the fold change in the PRNT_50_ before and after challenge suggested no difference among any of the groups.

### 3.3. RNA-Seq Analysis of Viral Genomic RNA Showed Greater Replication in Nasal Turbinates Compared to Lungs of Infected: Challenged Mice

To evaluate the potential for viral replication in the challenged mice, lungs and nasal turbinates were collected from four mice of each experimental group for RNA-Seq at 25 and 27 dpi (1 and 3 dpc) (Figure 1A). As controls for the high dose of infection, mice were infected with WA1, Alpha, or Delta, and at 1 and 3 dpi, lungs and nasal turbinates were collected for RNA-Seq (not shown on timeline, see Figure 3). These days were chosen as 3 dpi is the peak of infection for WA1 and 1 and 3 dpi provide insight into the early inflammatory response [15]. In this section, we present results regarding the measurement of viral RNA, and we present data from the RNA Seq analyses on viral genome SNPs in infected mice (Section 3.4) and the early inflammatory response (Section 3.5).

SARS-CoV-2 sequencing reads were graphed according to their depth of coverage across the genome for each sample (Figure 3). For comparison to the infected:challenged mice, we sequenced WA1, Alpha, and Delta at 1 and 3 dpi and assessed the viral genome levels in transcripts per million (TPM) as previously published by Phan et al., 2021 [19] using a similar bioinformatic pipeline. The viral RNA levels in the nasal turbinates and lungs of mice from the high dose of WA1 showed a high depth of coverage from 1 to 3 dpi, reflecting active replication (Figure 3A,B).

Following challenge at 1 and 3 dpc, viral RNA was at the limit of detection in the nasal turbinate and lung for the WA1:WA1 cohort (Figure 3A,B). The initial low-dose WA-1 infected mice were also evaluated at days 25 and 27 dpi and were below the limit of detection (Figure 3C). The limit of detection was set by reads detected in sham (PBS)-inoculated samples (Figure 3D).

As compared to the absence of reads in the WA1:WA1 cohort, viral RNA was detected in the heterologous infection:challenge groups, WA1:Alpha, and WA1:Delta (Figure 3A,B). WA1:Alpha reads were noted above the limit of detection at 1 and 3 dpc in the nasal turbinate whereas WA1:Delta reads were noted at 1 dpc (Figure 3A). In the lung, WA1:WA1 and WA1:Alpha were at or below the limit of detection (Figure 3B). A slight amount of reads were detected at 1 dpc in the WA1:Delta cohort in the lung, but not at 3 dpc (Figure 3B). The region spanning the coding region of the nucleocapsid (N) protein of the SARS-CoV-2 genome (nucleotides 28,274 to 29,533) had the greatest depth of coverage.

Viral RNA abundance by RNASeq was confirmed by using RT-qPCR to quantify the amount of SARS-CoV-2 in isolates from the lung (Figure 4A) and nasal turbinates (Figure 4B). Data from RT-qPCR experiments (represented as log copy number) supported findings from RNA-Seq data (transcripts per million, or TPM) in which breakout infection was observed in nasal turbinates from VOC-challenged mice (Figure 4B).

### 3.4. Nonsynonymous Mutations Emerged Early in Infection-Only Mice in Lungs and Nasal Turbinates

To assess the viral RNA genomes present in the NT and lung for low-level variants, SNPs were assessed against the reference genome of each virus. Detection of SNPs was limited to nucleotides of the genome that had a minimum read depth of 100 per position. To be included in our analysis of SNPs, the mutation had to be present in at least two samples and have greater than one percent frequency.

SNPs were identified in the WA1, Alpha, and Delta-infected mice and in one WA1:Alpha mouse, but not in the WA1:WA1 or WA1:Delta experimental groups (Appendix A). This finding does not imply that WA1 is more prone to accumulating mutations but may reflect the lower number of genomes in challenged mice, which impacts the limit of detection of SNPs. SNPs in WA1-infected mice were noted in the ORF1a and ORF1ab, S, M, and N regions of the SARS-CoV-2 genome (Appendix A). In the lung and nasal turbinate samples of mice infected by Delta, there were few SNPs (Appendix A). ORF8:Q23H was noted in lung and nasal turbinate at 1 and 3 dpi in WA1. The ORF1b:K2395N mutation was observed in WA1 lung samples at 3 dpi and nasal turbinate at 1 dpi. The S:E1072K mutation emerged in nasal turbinate samples of Delta-infected mice at 3 dpi. Evaluation of the genomes from the lungs and nasal turbinate of Alpha-infected mice revealed several SNPs, however, none were observed in more than one mouse.

There were several Ns mutations that stood out in the analyses. A mutation at M:T7I occurred in the lung and nasal turbinate of WA1-infected mice at 1 and 3 dpi (Figure 5A). Also, we noted a S:N215H mutation, which was observed with similar frequencies in lung and NT of WA1-infected mice at 1 and 3 dpi (Figure 5B). Interestingly, a N:S194T mutation emerged at a frequency of 98% in one sample from an Alpha-challenged mouse (Figure 5C), but this mutation was present at ~40% in our WA1 virus stock. The S:V62G was observed in WA1-infected mouse lungs at low frequency as well as in the lung of one Delta-infected mouse (Figure 5D). Additional studies are required to interpret the functionality of these amino acid changes.

### 3.5. A Dampened Proinflammatory Response in the Lung, but Not the Nasal Turbinate, of Heterologous-Challenged Mice

To evaluate the early immune response following challenge, the RNASeq data was analyzed for the upregulation of signaling pathways. To serve as controls, we included WA1-, Alpha-,or Delta-infected (2.5 × 10^4^ PFU) mice at 1 and 3 dpi. In the control mice, we noted upregulation of hypercytokinemia/hyperchemokinemia, phagosome formation, dendritic cell formation, and acute-phase response signaling pathways for nasal turbinates and lungs (Figure 6A,C). The acute phase response signaling was observed in the nasal turbinate (Figure 6A), but not the lung (Figure 6C). The level of viral RNA in nasal turbinates (Figure 6B) was lower than noted in lungs (Figure 6D) for all three, and WA1 had a slightly higher level of viral RNA than Alpha or Delta.

As compared to the infection-only groups (Figure 6A,C), WA:WA1, WA1:Alpha, and WA1:Delta had lower gene expression levels at 1 and 3 dpc (Figure 7A). In general, the greatest differences among the infected challenged groups were noted at 1 dpc; see WA1:WA1 and WA1:Delta groups in the nasal turbinate and also compare WA1:WA1 and WA1:Alpha cohorts in the lung (Figure 7B,C). In addition to hypercytokinemia pathways, the WA1:WA1 cohort was predicted to have a slight upregulation of the T cell receptor signaling pathway, LXR/RXR and neuroinflammatory pathways, and the downregulation of IL17 signaling (Figure 7A). The WA1:Delta mice showed the highest level of upregulation of IL17 signaling and LXR/RXR activation. The WA1:Delta group was unique in activation of intrinsic prothrombin activation, acute phase response signaling, production of nitric oxide and reactive oxygen species in macrophages, and the coagulation system. Lastly, the WA1:Alpha group of mice showed an upregulation of their Th2 and T cell exhaustion pathways. We evaluated each of the pathways using Venn diagrams (Figure 7B,C) to identify notable genes upregulated in the different groups (Appendix A).

We highlight our findings for some of the genes (Figure 8) that have been reported to be of relevance to the immune response during SARS-CoV-2 infection [20,21,22]. The expression of some of the genes in the hypercytokinemia pathway highlights the differences between the experimental groups (Figure 8, Appendix A). The proinflammatory response was largely absent from the lungs of the infected:challenged groups (Appendix A). Interestingly, CXCL10 and CXCL11 were elevated in all the infected and challenged groups of mice in the lung. In the nasal turbinate, CXCL10 and CXCL11 were absent in WA1:WA1 at 1 and 3 dpc as well as the challenged groups at 3 dpc (Appendix A). Proinflammatory genes that were upregulated in the WA1-infected group and dampened in the challenge groups included IFNG, CXCL10, IL6, and IFNB1 (Figure 8A). WA1-infected mice had a consistently higher fold increase at 1 and 3 dpi and WA1:WA1 had the lowest.

Some of the upregulated genes that contributed to the differential activation of the phagosome formation pathway (Figure 8B) included TLR7, TLR2, FCGR1A, and C3RA1. Challenged experimental groups of mice had a lower fold increase of these genes at 1 dpc and declined further at 3 dpc. Also notable, JChain and AICDA were only upregulated in lungs of the challenged groups (Appendix A).

## 4. Discussion

The presented SARS-CoV-2 infection:challenge study elucidated the protection conferred from a low-dose, live infection of WA1 to challenge with a high dose (2.5 × 10^4^ PFU) of homologous (WA1) or heterologous (Alpha, Delta) viruses. All mice were completely protected from death regardless of the challenge strain—WA1, Alpha, or Delta. However, the recovery of the mice that were challenged with the heterologous variants of concern lagged that of the mice challenged with the homologous infectious dose. WA1:Alpha and the WA1:Delta cohorts resumed weight gain after three to four days of decline. None of the challenged mice showed any clinical symptoms outside of the weight loss for three weeks from the date of challenge. A comparison of the fold change of the PRNT_50_ (before 21 dpi and after challenge at 21 dpc) suggested that all challenge groups had a similar boost in neutralizing titer. The similarity of the PRNT_50_ titers for WA1:WA1, WA1:Alpha, and WA1:Delta cohorts suggests the boost to the neutralization titer was likely due to WA1 shared epitopes and very little contributions from Alpha- or Delta-specific epitopes.

Viral replication was confirmed in WA1:Alpha and the WA1:Delta cohorts but was at or below the limit of detection in WA1:WA1 mice based on RNASeq and RT-qPCR data from sham-infected:challenged (PBS:PBS) control groups (Figure 3 and Figure 4). Hence these data show no viral RNA remaining for the WA1:WA1 cohort at 1 and 3 dpc or the WA1-infected cohort at 25 and 27 dpi (Figure 3 and Figure 4). In contrast, replication of viruses in the WA1:Alpha and WA1:Delta groups was noted at the subgenomic (3′) end (Figure 3). This can be attributed to the SARS-CoV-2 discontinuous mechanism of transcription, which results in abundant subgenomic transcript generation [23,24]. In spite of the similar PRNT_50_ findings, our results suggest that the challenge VOCs were cleared less effectively in the nasal turbinates and lung tissues by the NAbs produced from the initial WA1 infection. Since T cell analysis was not included in our study, we are unable to comment on its role in conferring protection, but there is ample evidence that describes its importance to cellular immunity [25]. Similar protective efficacy has been shown in SARS-CoV-2 infection:challenge studies using mice, ferrets, hamsters, and rhesus macaques [26,27,28,29,30,31,32,33]. A mild infection of SARS-CoV-2 in the K18-hACE2 mouse model has previously been shown to be protective against a high dose of reinfection with the same virus for up to 24 weeks, with mice reinfected 24 weeks later showing some weight loss [27]. This study also reported no infectious virus in the lungs, but they did not evaluate the virus or host response in the upper respiratory tract. Similar studies in mice have evaluated SARS-CoV-2 vaccination–infection dynamics with most of the studies evaluating a single virus in the challenge [26,29,31]. However, in the comprehensive assessment of WA1 and Beta mRNA vaccination in the K18-hACE2 transgenic mice with homologous and heterologous infection (WA1, Alpha, Beta, Delta), a high dose of vaccination provided protection with no evidence of breakthrough of infectious virus in nasal washes, lungs or brain [32]. Hamsters infected with WA1 show protection from reinfection with WA1 for up to 4 months, but are able to transmit virus to cohoused naïve hamsters [30]. In the same study, reinfection of WA1-infected hamsters with Beta showed breakthrough levels of virus in the nasal wash but not the lung [30]. The rhesus macaque model does not recapitulate the disease severity or lethality of SARS-CoV-2 noted in hamsters or mice, respectively; however, in these studies, the WA1-infected macaques mounted an effective immune response to WA1 following challenge that exceeded that of the infected group, but virus was detected in the nasal wash samples [28]. Hence, while great progress has been made, a gap remains in our understanding of protection dynamics from heterologous reinfection in the nasal turbinates. Additionally, we lack insight into mechanisms promoting the emergence of nonsynonymous mutations in infection–reinfection studies that would provide insight into how the virus might evade the immune response in the upper versus the lower respiratory tract. Interestingly, herein we observed nonsynonymous mutations in the naïve-infected mice early after infection, which suggests the importance of the evaluation of the genetic plasticity of the virus and the resulting functionality of the mutation. For example, we noted the V62G, which is a mutation in the N terminal domain of spike, a site with immunogenic activity [34].

An early elevation in the proinflammatory response of the WA1:Alpha and the WA1:Delta cohorts (Figure 7), but not WA1:WA1, was observed that corresponded to the period of weight loss following challenge (Figure 1); however, the host responses at 1 and 3 dpc (Figure 7) were greatly reduced as compared to mice infected with WA1, Delta, or Alpha at 1 and 3 dpi (Figure 6). Of note, at 1 dpc, the IFNβ and IFNγ RNA levels were higher in nasal turbinates of WA1:Alpha and WA1:Delta but were low or near baseline in the lung. Similar results were observed for other cytokines and chemokines such as IL6, IRF7, OAS2, and OAS3 (Appendix A) but not for CXCL10, which was upregulated in the lung for WA1-infected and all challenge groups. CXCL10 is critical for CD8+ and CD4+ T cell recruitment and has been demonstrated to correlate with the severity of COVID-19 [35,36,37]. Our findings for the WA1:WA1 cohort are similar to that reported for SARS-CoV in which infection followed by homologous challenge in ferrets results in an anamnestic response in the lung that abrogated infection without reinitiating acute inflammation [38]. TLR7 functions in the intracellular sensing of single-stranded RNA viruses [39] and as anticipated were upregulated in the WA1-infected mice, but were mostly unchanged in the WA1:Alpha and the WA1:Delta groups. These data show that inflammation of the upper respiratory tract coincided with active infection.

In addition to proinflammatory signaling, several other canonical pathways associated with the challenges were detected. As noted in the proinflammatory signaling pathway and in the phagosome formation pathway, more genes were upregulated in the nasal turbinates than in the lungs of challenged groups. For example, FCGR1A, Fc gamma receptor 1a, or CD64, which is expressed in antigen-presenting cells and mediates binding to the Fc region of antibody to combat infection and drive inflammation [40], was upregulated in lungs of WA1-, Alpha-, and Delta-infected mice 1 and 3 dpi, but not in the challenged groups. C3RA1, the receptor for C3a, which functions in the complement system of the innate immune response and triggers the expression of downstream proinflammatory genes in response to SARS-CoV-2 infection [41], was upregulated 3 dpi in the lungs of WA1-, Alpha-, or Delta-infected mice. C3RA1 was also upregulated in nasal turbinate of mice infected with WA1 3 dpi. Since the local immune response of the nasal turbinate is seldom characterized in vaccine studies, it is unclear how these results would differ between vaccinated and naturally infected hosts.

As SARS-CoV-2 variants continue to emerge with the potential to spread globally, more studies are needed to fully characterize how viral infection–reinfection, vaccination–infection and infection–vaccination dynamics impact B cell generation and expansion and diversification of antibody populations to better assess risk for patients and inform immunization schedules. While few studies have reported the associated level of protection of natural immunity conferred following infection, at least one study with 9 to 19 months of follow up suggests long-lasting protection against severe COVID-19, although this protection wanes faster in older adults [42,43]. Hence, study of naturally acquired immunity may provide new insights into vaccine strategies. The ability of vaccines to prevent against severe COVID-19 by protecting the lung has been studied extensively; however, protection in the upper respiratory tract has not been characterized to nearly the same degree. Our findings in the upper respiratory tract underscore the translational impact of this work and the need for the development of vaccines that confer mucosal immunity as an additional form of immunization. In conclusion, future studies that address the mechanistic differences in how emerging SARS-CoV-2 variants evade and modulate inflammation in natural and vaccinated individuals are critical to development of optimal therapeutic and vaccine strategies.

## Figures and Tables

**Figure 1 viruses-15-00946-f001:**
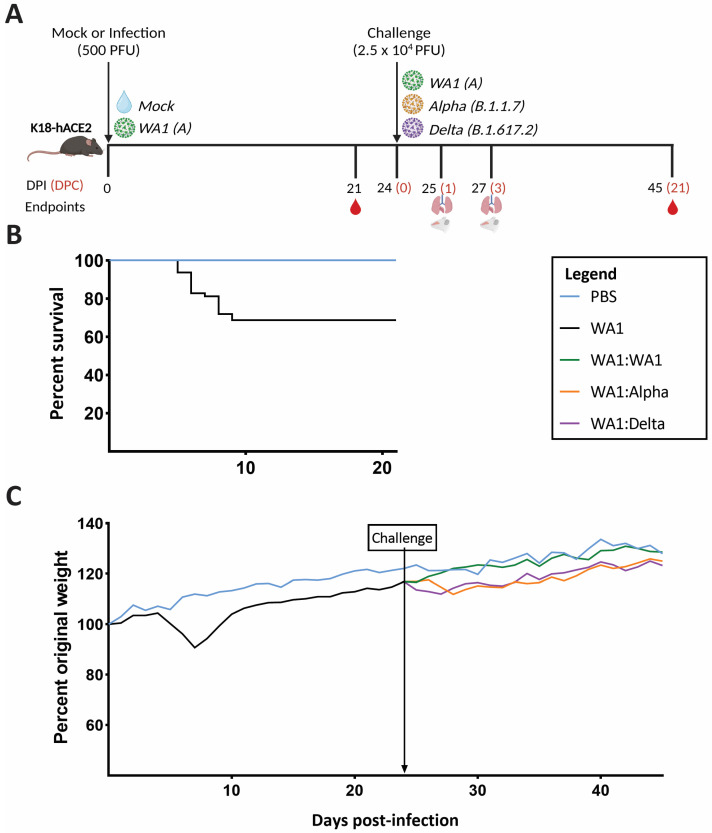
Study timeline and clinical outcomes. (**A**) Mice (n = 104) were inoculated intranasally with either PBS (sham) (n = 40) or 500 PFU of WA1 (n = 64). At 24 days postinfection (dpi) or 0 days postchallenge (dpc), WA1-challenged mice were inoculated with 2.5 × 10^4^ PFU of WA1 (n = 12), Alpha (n = 12), or Delta (n = 12) (noted by “Challenge” and arrow). In addition, 24 mice were infected for the first time from the PBS control group with WA1 (n = 8), Alpha (n = 8), or Delta (n = 8) or PBS (n = 8). At 25 (1 dpc) and 27 (3 dpc) dpi, lungs and/or nasal turbinates were collected for RNA-Seq. At 21 and 45 dpi, blood was collected to test neutralizing titers. The figure was created with Biorender.com. (**B**) The survival graph of mice infected with WA1 (n = 64) shows a 31.3% mortality. (**C**) For clarity, the average of the percent of original weight at 0 dpi for the 5 groups of mice is shown. Prior to and through 24 dpi, PBS (blue line) and WA1 (black line) inoculated mice are shown. Beginning at 25 dpi, the average percent of weight change of the WA1 infection plus the challenge virus are shown as green (WA1), orange (Alpha), and purple (Delta) lines. The blue line shows the average weight change of PBS. Nonlinear regression modeling of each group’s average percent weight change graph between 24 and 28 dpi showed a slope of 1.08 for WA1-challenged mice and a slope of −0.67 and −1.44 for Alpha- and Delta-challenged mice, respectively. A one-way ANOVA showed *p* values of 0.0129 and 0.0199 for the groups WA1:Alpha and WA1:Delta, respectively, suggesting a significant difference between the homologous- and heterologous-challenged groups. Abbreviations: dpi—days postinfection; dpc—days postchallenge.

**Figure 2 viruses-15-00946-f002:**
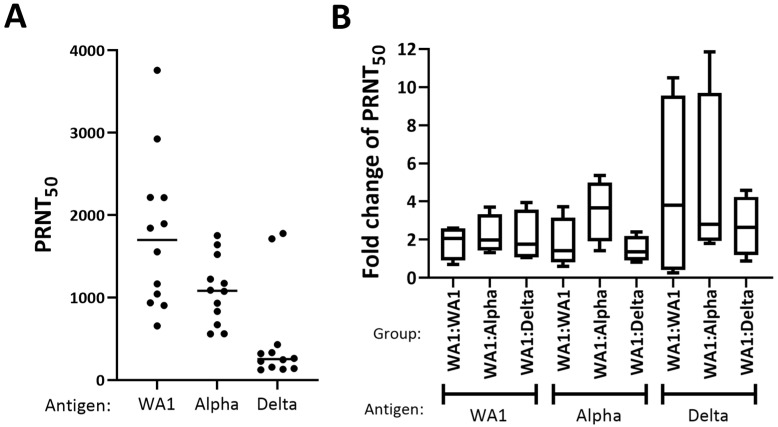
Plaque reduction neutralization titers (PRNT_50_) prior to challenge and PRNT_50_ fold change after challenge across three viral antigens. On the X-axis for each panel, the virus (i.e., antigen) used to measure the PRNT_50_ is shown for each experimental group. (**A**) Fifty percent plaque reduction neutralization titers prior to homologous and heterologous challenge. (**B**) The fold changes in the PRNT_50_ values were calculated for serum sample collected at 21 days postinfection (dpi) and 45 dpi (21 days postchallenge). The geometric mean is shown in each box plot as a dark line. The fold changes of PRNT_50_ between 21 dpc and 21 dpi were log-transformed and to measure significance, we used a one-sided, one-sample *t*-test to test a group mean greater than zero or fold change greater than one. We also used a two-sample *t*-test to compare the difference in fold changes between the two groups. We used robust linear regression to model fold change (log-transformed) with rechallenge and antigen as predictors. We did not observe a significant difference among challenge groups with any antigen used to measure PRNT_50_ (*p* value = 0.39).

**Figure 3 viruses-15-00946-f003:**
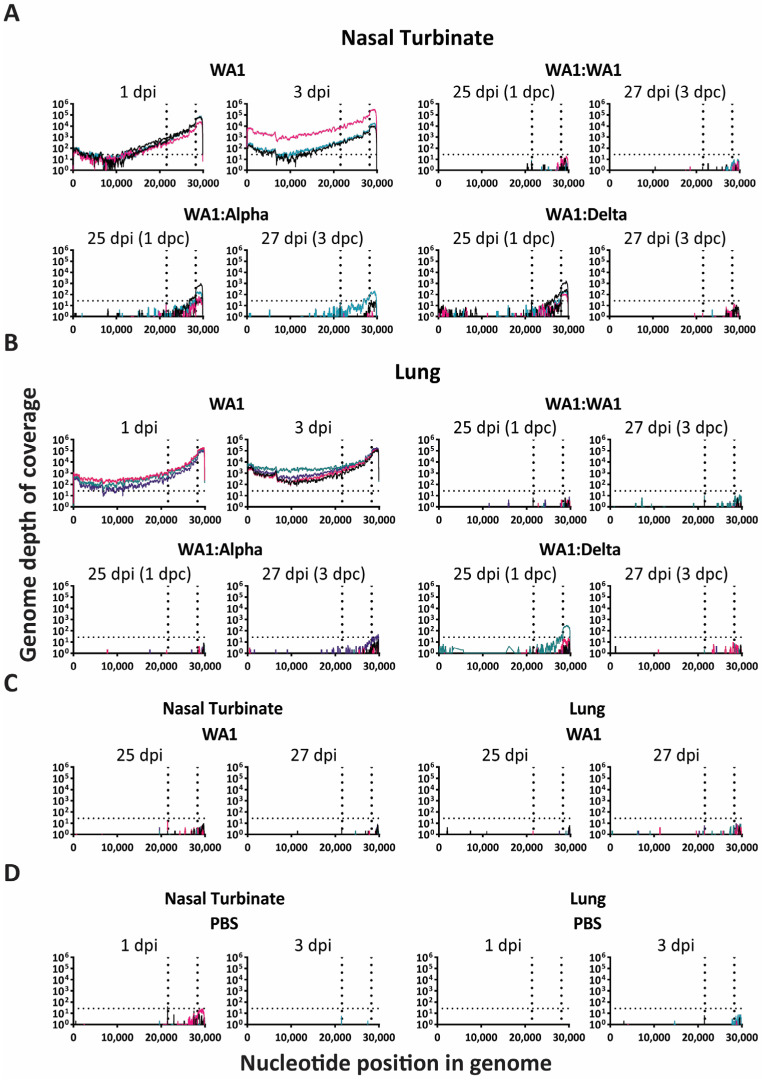
Viral RNA in lungs and nasal turbinates of challenged mice is limited to subgenomic transcripts. RNA-seq data of the viral genome were graphed to show depth of coverage (Y-axis) by nucleotide position (X-axis) on the SARS-CoV-2 genome for nasal turbinate (**A**,**C**,**D**) and lung (**B**–**D**) samples. The first vertical dotted line marks the beginning of the subgenomic region (the start of the spike) and the second vertical dotted line marks the beginning of the nucleocapsid encoding region. The horizontal dotted line represents the limit of detection. Abbreviations: dpi—days post-infection; dpc—days postchallenge. Each colored line represents the results from one mouse.

**Figure 4 viruses-15-00946-f004:**
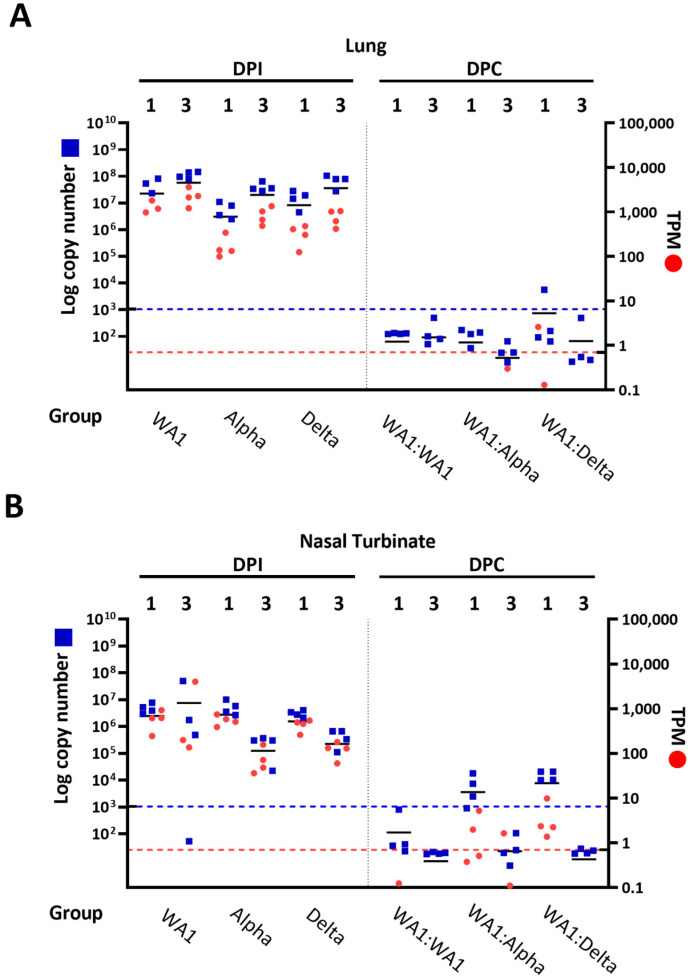
RT-qPCR data corroborate results from RNA-Seq to quantify virus recovered from lungs (**A**) and nasal turbinates (**B**) of mice. Log copy number of SARS-CoV-2 (left Y-axis) is overlayed with the transcripts per million (TPM) (right Y-axis). Groups are listed along the bottom X-axis based on their treatment and the day of organ collection is listed horizontally across the top of the graph. The blue horizontal dotted line serves as the lower threshold and marks the highest log copy number recorded in a sham-inoculated mouse. The red horizontal dotted line serves as the lower threshold of TPM and marks the highest TPM count in a sham-infected mouse. The vertical black dotted line in the middle of the graph separates the mouse groups with the infected on the left and the infected:challenged on the right. Abbreviations: dpi—days post-infection; dpc—days postchallenge.

**Figure 5 viruses-15-00946-f005:**
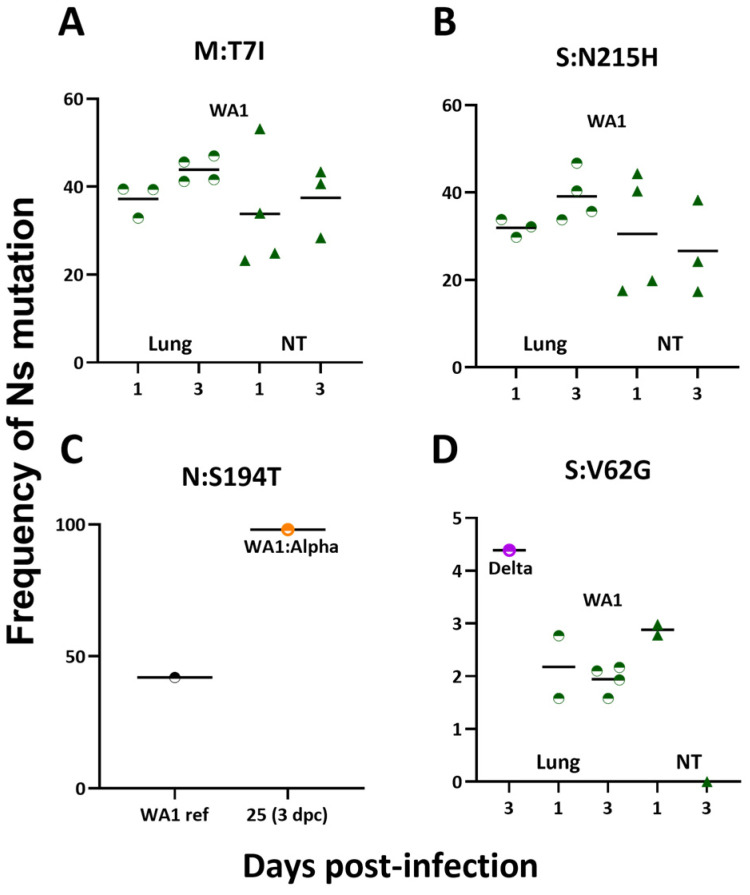
Nonsynonymous mutations in lungs and/or nasal turbinates at 1 and 3 days postinfection or postchallenge. The frequencies of nonsynonymous mutations that were noted in the lung and nasal turbinate (NT) for WA1-infected mice (**A**,**B**,**D**), Delta-infected mice (**D**), and WA1-infected:Alpha-challenged mice (**C**). (**A**) M:T7I was detected in WA1-infected mice 1 and 3 days postinfection (dpi) in the lung and nasal turbinate (NT). (**B**) S:N215H was detected in WA1-infected mice 1 and 3 dpi in lung and nasal turbinate. (**C**) N:S194T was detected in our WA1 stock at just under 50% as well as in one Alpha-challenged mouse at just under 100% frequency. (**D**) S:V62G was found in lungs and NT of WA1-mice 1 and 3 dpi and in the lung of a Delta-infected mouse 3 dpi. Abbreviations: ref—reference; NT—nasal turbinate; dpi—days postinfection; dpc—days postchallenge.

**Figure 6 viruses-15-00946-f006:**
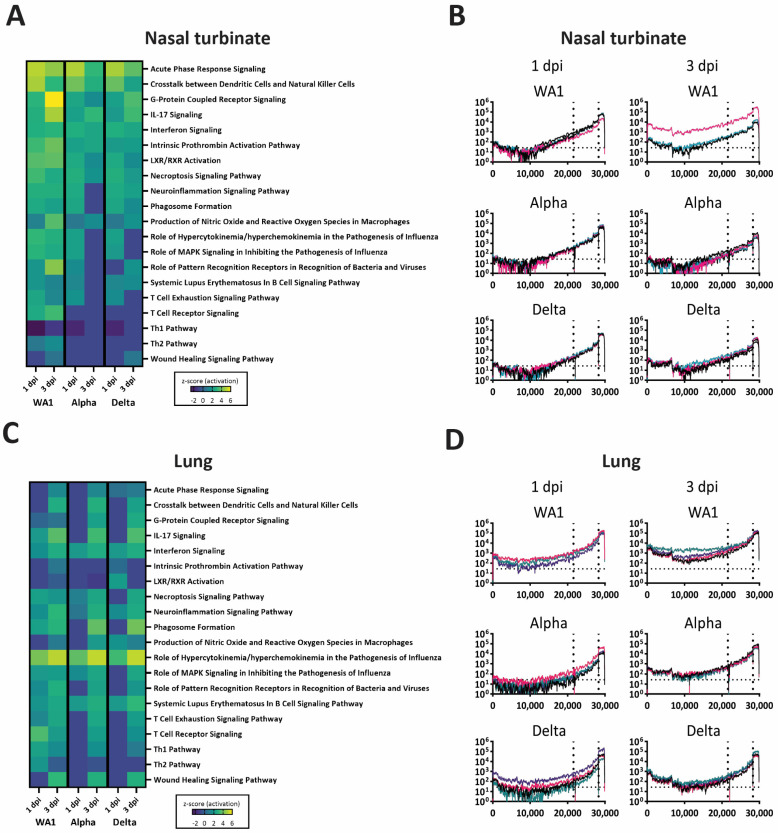
Heatmap of activated pathways and viral RNA levels in nasal turbinates and lungs from high dose of WA1-, Alpha-, or Delta-infected mice. (**A**,**C**) Heatmap illustration of the activated pathways in mice infected with 2.5 *×* 10^4^ PFU of WA1, Alpha, or Delta as compared to PBS-inoculated at 1 and 3 days postinfection (dpi). Canonical pathways are displayed on the right and group specifications are listed below. The dpi is listed at the bottom of each heat map. As described in the Materials and Methods section, the reads from RNASeq data were mapped to *Mus musculus* and the SARS-CoV-2, gene counts were assessed in DESeq2, and only those genes with a log_2_ fold change of ±1.5 with an FDR ≤ 0.05 were evaluated in Ingenuity Pathway Analyses software. RNA-seq data of each viral genome were graphed to show the depth of coverage (Y-axis) by nucleotide position (X-axis) on the SARS-CoV-2 genome for nasal turbinate (**B**) and lung (**D**) samples. In each graph, the vertical dotted line marks the beginning of the subgenomic region, the start of the spike encoding region. Second vertical dotted line marks the beginning of the nucleocapsid encoding region. The horizontal dotted line represents the limit of detection. Legend: Z-scores are from dark purple (lowered) to yellow (heightened), as illustrated. Each colored line represents the results from one mouse.

**Figure 7 viruses-15-00946-f007:**
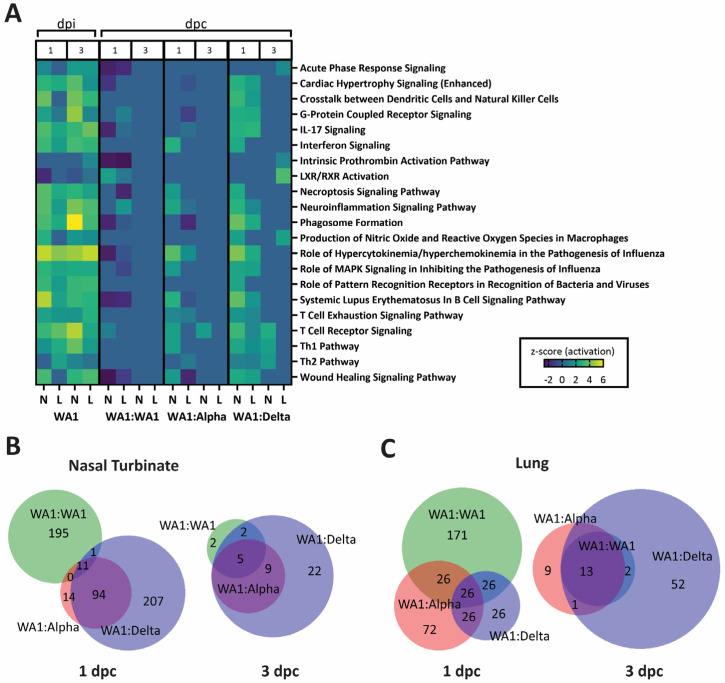
Heatmap of activated pathways in nasal turbinates and lungs from WA1-infected or WA1:WA1-, WA1:Alpha-, and WA1:delta-challenged mice. (**A**) Heatmap illustration of the activated pathways in mice infected with 2.5 *×* 10^4^ PFU of WA1 or challenged with 2.5 *×* 10^4^ PFU of WA1, Alpha, or Delta) as compared to PBS-inoculated at 1 and 3 days postchallenge (dpc). Canonical pathways are displayed on the right and group specifications are listed below. Days postinfection (dpi) or dpc are listed at top. As described in the Materials and Methods section, the reads from RNASeq data were mapped to Mus musculus and the SARS-CoV-2, gene counts were assessed in DESeq2, and only those genes with a log_2_ fold change of ±1.5 with an FDR ≤ 0.05 were evaluated in Ingenuity Pathway Analyses software. Venn diagrams present the number of unique and shared genes between groups in nasal turbinates (**B**) and lungs (**C**). Legend: Z-scores are from dark purple (lowered) to yellow (heightened), as illustrated. Abbreviations: days postinfection (dpi); days postchallenge (dpc), L—lung, N—nasal turbinate.

**Figure 8 viruses-15-00946-f008:**
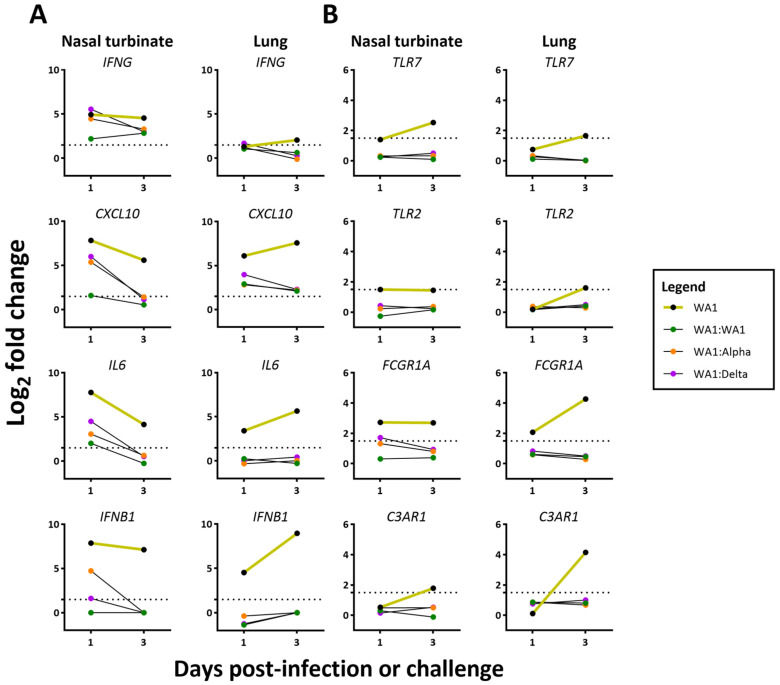
Differential expression of select genes from lung and nasal turbinate samples from WA-1 infected and challenged mice. Graphs comparing log_2_ fold change of specific genes in WA1-infected (yellow), WA1:WA1- (green), WA1:Alpha- (red), and WA1:Delta-challenged (purple) groups that are integral to the (**A**) hypercytokinemia–hyperchemokinemia and (**B**) phagosome formation pathways in nasal turbinate and lung samples from 1 and 3 days postinfection from WA1-infected and 1 and 3 days postchallenge for WA1:WA1, WA1:Alpha, and WA1:Delta. As described in the Materials and Methods section, the reads from RNASeq data were mapped to Mus musculus and the SARS-CoV-2, gene counts were assessed in DESeq2. The dotted line indicates a log_2_ fold change of 1.5. All genes have a log_2_ fold change above 1.5 and an FDR ≤ 0.05, except for NT WA1-TLR2 (0.47), TLR7 (0.25), NT WA1:WA1-IFNG (0.21), CXCL10 (0.27), IL6 (0.33), NT WA1:Alpha-IL6 (0.27), IFNB1 (0.28), NT WA1:Delta-IFNB1 (0.83), FCGR1A (0.13), and IFNG for all 3 dpc NT samples.

## Data Availability

All data are available in the main text or the Appendix A. Consensus sequence for UT2590 has been deposited at Genbank (Accession no. OP628428). RNASeq files have been deposited at the NIH Gene Expression Omnibus (GSE216048) http://www.ncbi.nlm.nih.gov/geo (accessed on 13 October 2022).

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
