# Peer review of "Upper Respiratory Infection Drives Clinical Signs and Inflammatory Responses Following Heterologous Challenge of SARS-CoV-2 Variants of Concern in K18 Mice"

_viruses, 2023, doi:10.3390/v15040946_

Round 1
Reviewer 1 Report
Jacob and collaborators described “Upper respiratory infection drives clinical signs and inflammatory responses in heterologous challenge of SARS-CoV-2 variants of concern”. The paper was well written and tried to answer an extremely important question, what drives the clinical symptoms of re-infection? However, the data in current manuscript were not enough to support authors’ claim that “self-limiting breakthrough infection of Alpha or Delta in the URT which correlated with clinical signs and a significant inflammatory response in mice” (line 29-30).
In Figure 1B, the data showed a 31.5% mortality with a low dose (500PFU) infection, which raises the question of why this so-called low dose actually caused high mortality, and of whether the survived mice developed extremely high neutralizing antibodies, which is missing in the current manuscript.
In Line 194, the authors said no other clinical symptoms were observed, of which the data were missing in the manuscript. It would be good to clarify what investigations were performed.
All the figures in the current manuscript lacked statistical analysis labels. In Figure 2, the data showed no difference in the fold change of PRNT50 between any groups. However, the absolute neutralizing antibody titer is critical for effective protection, which was missing in current manuscript.
Authors used RNAseq data to show greater replication in nasal turbinates (line224). However, this is not a right way to confirm the active viral replication. The following-up RT-qPCR assay can only validate the RNA-seq data.
Could the authors clarify the connection between section 3.4 (Line267-306) and the topic of the current study?
In section 3.5, the dpi was misused (line 310, Fig6, B and D). And some functional assays are needed to support the correlation of self-limiting breakthrough infection of Alpha or Delta in the URT and inflammatory response.
Author Response
please see attached as we included a figure
Jacob and collaborators described “Upper respiratory infection drives clinical signs and inflammatory responses in heterologous challenge of SARS-CoV-2 variants of concern”. The paper was well written and tried to answer an extremely important question, what drives the clinical symptoms of re-infection?
- However, the data in current manuscript were not enough to support authors’ claim that “self-limiting breakthrough infection of Alpha or Delta in the URT which correlated with clinical signs and a significant inflammatory response in mice” (line 29-30).
Response: We appreciate this concern in our using this newer approach to evaluate viral replication by RNASeq and we have now included the reference for the premise of our approach and soften the conclusion. We based our RNASeq approach to quantify viral RNA levels on the publication by Phan T, Fay EJ, Lee Z, Aron S, Hu WS, Langlois RA. Segment-specific kinetics of mRNA, cRNA and vRNA accumulation during influenza infection. J Virol. 2021 Apr 26;95(10):e02102-20. PMID: 33658346. In this publication the authors validate the use of RNASeq for quantitative measurement of viral RNAs and our approach was identical. They argue that the use of the RNASeq TPM measurement removes primer bias and overcomes technical hurdles associated with qRT-PCR. In the conclusion, we have modified our wording to suggested: suggested a self-limiting breakthrough infection of Alpha or Delta in the URT which correlated with clinical signs and a significant inflammatory response in mice.
- In Figure 1B, the data showed a 31.5% mortality with a low dose (500PFU) infection, which raises the question of why this so-called low dose actually caused high mortality, and of whether the survived mice developed extremely high neutralizing antibodies, which is missing in the current manuscript.
Response: We appreciate the question regarding the lethal dose of the WA1 strain. The dose was chosen based on our LD50 studies to ensure a sufficiently strong dose that would allow survival for the challenge. We have added the following sentence: The dose was chosen as to enable infection but allow at least two-thirds of the cohort to survive for the subsequent challenge reported previously (PMID: 35634338).
We have noted over the past three years that there is lethality at 500 PFU and below.
As mentioned above, we have another published paper (PMID: 35634338) using an even lower dose, 250PFU, in which mice infected with SARS-CoV-2 was lethal in 35% of mice. So, this was known beforehand. The neutralizing titers of the WA1 only challenge are presented in the new Figure 2A.
- In Line 194, the authors said no other clinical symptoms were observed, of which the data were missing in the manuscript. It would be good to clarify what investigations were performed.
Response: Thank you for pointing this out. We have added this sentence to the manuscript at the end of section 3.1 and in section 2.2. The clinical signs that we monitored included lethargy based on appearance, respiratory effort, weight loss, body temperature reduction, behavior (activity), and hydration status.
- All the figures in the current manuscript lacked statistical analysis labels. In Figure 2, the data showed no difference in the fold change of PRNT50 between any groups. However, the absolute neutralizing antibody titer is critical for effective protection, which was missing in current manuscript.
Response: We appreciate this suggestion and we have added a figure of the neutralizing titers of the WA1 infected mice using WA1, Alpha or Delta as the PRNT antigen (new Figure 2A). We also add the raw data as a supplement for each mouse (Supplemental Table 4). We have also added the statistical information for each figure in the figure legend as appropriate.
- Authors used RNAseq data to show greater replication in nasal turbinates (line224). However, this is not a right way to confirm the active viral replication. The following-up RT-qPCR assay can only validate the RNA-seq data.
Response: We appreciate this as this is a new approach (see number 1 above and Phan T, Fay EJ, Lee Z, Aron S, Hu WS, Langlois RA. Segment-specific kinetics of mRNA, cRNA and vRNA accumulation during influenza infection. J Virol. 2021 Apr 26;95(10):e02102-20. PMID: 33658346) and that is why we have included the qRTPCR data as addition support for virus replication.
- Could the authors clarify the connection between section 3.4 (Line 267-306) and the topic of the current study?
Response: We apologize for the lack of clarity on our objective, and we have edited this to clarify this in the text.
- In section 3.5, the dpi was misused (line 310, Fig6, B and D). And some functional assays are needed to support the correlation of self-limiting breakthrough infection of Alpha or Delta in the URT and inflammatory response.
Response: We double checked and the term dpi is correct in this statement. In this section, we are only reporting the variants noted in at least two or more mice. We reviewed and edited the text to make sure that we are not suggesting these are functional.

Reviewer 2 Report
The original article by Nichols and co-authors is very well-written and novel. The data presented is very clear and supports the conclusions. There is only a minor comment regarding the figures. The authors should define the statistical method used for the figure in each corresponding legend. It will help readers to read it.
Otherwise, the research is of a very high standard and the manuscript should definitely be accepted for publication.
Author Response
The original article by Nichols and co-authors is very well-written and novel. The data presented is very clear and supports the conclusions.
- There is only a minor comment regarding the figures. The authors should define the statistical method used for the figure in each corresponding legend. It will help readers to read it. Otherwise, the research is of a very high standard and the manuscript should definitely be accepted for publication.
Response: Thank you for this suggestion. We have also added the statistical information for each figure in the figure legend or in the figure as appropriate.
Reviewer 3 Report
The authors performed SARS-CoV-2 infection and challenge experiment in K18-hACE2 transgenic mice. They first infected mice with low dose-WA1 then challenged on 24 dpi with homologous or heterologous Strains. This is a quite interesting animal model to investigate the role of re-infection or breakthrough infection. Comments for the authors:
Major points:
1. Please include (n) in Figure 1A. It is not clear how many mice were first infected with WA1. Line 109 (n=68) and Line 182 (n=64).
2. Please explain what happed to 40 mock infected mice. Did authors also challenge them? The authors should show the result of high-dose challenge without low-dose prime for comparison.
3. Lin 201; mortality should be 31.3% (20/64).
4. Please include error bars in Figure 1C.
Minor points:
1. Figure 6; ‘dpi’ should be ‘dpc’? Please make sure to use correct abbreviation throughout the manuscript.
Author Response
Please see attached as we added a figure
The authors performed SARS-CoV-2 infection and challenge experiment in K18-hACE2 transgenic mice. They first infected mice with low dose-WA1 then challenged on 24 dpi with homologous or heterologous Strains. This is a quite interesting animal model to investigate the role of re-infection or breakthrough infection. Comments for the authors:
Major points:
- Please include (n) in Figure 1A. It is not clear how many mice were first infected with WA1. Line 109 (n=68) and Line 182 (n=64).
Response: We really appreciate the reviewer noticing this; this was an oversight in our editing. There were 64 and this is updated in the text.
- Please explain what happed to 40 mock infected mice. Did authors also challenge them? The authors should show the result of high-dose challenge without low-dose prime for comparison.
Response: We have rewritten the figure 1 legend and we noted the issue in our review. We apologize for the lack of clarity and we have hopefully improved the clarity of the information.
- Lin 201; mortality should be 31.3% (20/64).
Response: We really appreciate the reviewer noticing this. We clearly meant to write 31.25. this has been edited as suggested.
- Please include error bars in Figure 1C.
Response: We have created a figure that includes the error bars for each separately. We show below the graphs for each infection as the figure becomes quite difficult to view when all the weight data is added with the deviations. In addition, we have clarified in the text that we do not wish to state that the overall body weights were significantly different. We only wish to state that after challenge the slopes of the average weight differed for WA1:WA1, WA1:Alpha and WA1:Delta (red bar in graphs below). Please let us know if this information should be added to the supplement.
Minor points:
- Figure 6; ‘dpi’ should be ‘dpc’? Please make sure to use correct abbreviation throughout the manuscript.
Response: Figure 6 are our controls so these are actually dpi. We have also rechecked the manuscript as suggested.

Round 2
Reviewer 1 Report
The authors addressed appropriately all issues I raised.